# The Pathophysiology of Collateral Circulation in Acute Ischemic Stroke

**DOI:** 10.3390/diagnostics13142425

**Published:** 2023-07-20

**Authors:** Marilena Mangiardi, Adriano Bonura, Gianmarco Iaccarino, Michele Alessiani, Maria Cristina Bravi, Domenica Crupi, Francesca Romana Pezzella, Sebastiano Fabiano, Enrico Pampana, Francesco Stilo, Guido Alfano, Sabrina Anticoli

**Affiliations:** 1Department of Stroke Unit, San Camillo-Forlanini Hospital, 00152 Rome, Italy; 2Unit of Neurology, Neurophysiology, Neurobiology, Department of Medicine, Campus Bio-Medico University, 00128 Rome, Italy; adriano.bonura@gmail.com (A.B.); gianmarco.iaccarino@gmail.com (G.I.); michelealessiani@gmail.com (M.A.); mcbravi@tiscali.it (M.C.B.); domcrupi@gmail.com (D.C.); frpezzella@gmail.com (F.R.P.); sabrina.anticoli@gmail.com (S.A.); 3Department of Neuroradiology and Interventional Neuroradiology, San Camillo-Forlanini Hospital, 00152 Rome, Italy; sebastianofabiano@gmail.com (S.F.); enricopampana@hotmail.com (E.P.); 4Unit of Vascular Surgery, Campus Bio-Medico University, 00128 Rome, Italy; f.stilo@policlinicocampus.it; 5Department of Radiology and Interventional Radiology, M.G. Vannini Hospital, 00177 Rome, Italy; guidoalfano@yahoo.it

**Keywords:** acute ischemic stroke, collateral circulations, penumbra, thrombectomy, intravenous thrombolysis, anterior brain circulation, posterior brain circulation

## Abstract

Cerebral collateral circulation is a network of blood vessels which stabilizes blood flow and maintains cerebral perfusion whenever the main arteries fail to provide an adequate blood supply, as happens in ischemic stroke. These arterial networks are able to divert blood flow to hypoperfused cerebral areas. The extent of the collateral circulation determines the volume of the salvageable tissue, the so-called “*penumbra*”. Clinically, this is associated with greater efficacy of reperfusion therapies (thrombolysis and thrombectomy) in terms of better short- and long-term functional outcomes, lower incidence of hemorrhagic transformation and of malignant oedema, and smaller cerebral infarctions. Recent advancements in brain imaging techniques (CT and MRI) allow us to study these anastomotic networks in detail and increase the likelihood of making effective therapeutic choices. In this narrative review we will investigate the pathophysiology, the clinical aspects, and the possible diagnostic and therapeutic role of collateral circulation in acute ischemic stroke.

## 1. Introduction

In recent times, researchers have bestowed significant attention upon the role of collateral circulation in acute ischemic stroke due to its correlation with patient outcomes. It has been demonstrated that stroke patients with inadequate collateral circulation experience a four-fold higher rate of deterioration compared to others [1]. The presence of robust collateral networks can be considered an independent predictor of improved functional outcomes at the 90-day mark [2].

The significance of collateral circulation in the brain stems from the core/penumbra concept, which refers to the observation that a gradient of hypoperfusion is observed during acute ischemic stroke. This gradient ranges from a central region of severe hypoperfusion to peripheral areas with mild hypoperfusion. Visualizing this phenomenon, three concentric zones can be envisioned: a “core” region characterized by severe hypoperfusion, a “penumbra” area where neural cells are dysfunctional yet salvageable with adequate perfusion restoration through intravenous thrombolysis (IVT) and/or mechanical thrombectomy (MT), and an “oligemia” zone where hypoperfusion is mild and recovery is expected even without treatment.

Notably, it has been observed that tissue subjected to severe hypoperfusion has a shorter survival period, and the penumbra tissue will inevitably progress to irreversible damage if perfusion is not restored. The collateral circulation determines the fate of the penumbra, although it cannot indefinitely salvage it [3]. An inverse correlation exists among total infarct volume, collateral grade (indicating the extent of vascularity), and clinical outcome (assessed via the modified Rankin scale at discharge). Due to variations in vascular arrangement, collaterals afford greater protection to the cortex compared to deep tissues [4]. Additionally, for every 10 mL increase in pretreatment ischemic core volume, the odds of achieving a favorable functional outcome diminish by 20–30% [5].

The recognition that the core expands over time in the absence of restored perfusion has had a profound impact, leading to the development of revascularization procedures that are currently the only therapeutic strategies capable of altering the prognosis of stroke patients. The timing of intervention directly influences the impact on functional outcomes and the reduction of associated complications, such as hemorrhagic transformation [6]. Indeed, the most recent guidelines recommend thrombolysis within 4.5 h without advanced imaging (CT perfusion) or within 9 h with CT perfusion, as well as thrombectomy within 6 h without CT perfusion or within 24 h if CT perfusion is available [7]. Although IVT is currently recommended prior to MT, there is a debate about whether IVT is necessary for patients directly arriving in the stroke center with thrombectomy expertise. A recently published trial (DIRECT-MT) suggested that direct MT alone was non-inferior to MT preceded by IVT administered within 4.5 h after symptom onset. However, the trial had a liberal noninferiority margin (20%); a long onset-to-IVT time (184 min); and very short delay from start of IVT to groin puncture [8]. Mismatch between CT and CT Perfusion (CTP), or between diffusion-weighted and perfusion-weighted MRI (DWI and PWI) may quantify the penumbral cerebral tissue and could identify patients who benefit of alteplase beyond 4.5 h. Furthermore, one must differentiate between strokes occurring after 4.5 h and strokes upon awakening. 

However, despite timely and appropriate administration of these therapies, optimal functional outcomes are not always achieved. Several studies have acknowledged the crucial prognostic role of collateral circulation, which proves useful in both the diagnostic phase for guiding therapeutic decisions and the post-revascularization phase for prognostic purposes.

## 2. Anatomy and Neurophysiology

### 2.1. The Anatomy of Collateral Anterior and Posterior Circulation

The cerebral collateral circulation denotes the intricate network of blood vessels responsible for supplying the brain parenchyma with an adequate blood and oxygen supply in situations where the main arteries fail to function properly.

Within the anterior circulation, a distinction can be made between primary collaterals, represented by the components comprising the circle of Willis, and secondary collaterals, encompassing the ophthalmic artery and leptomeningeal vessels. Collateral vessels can be established between intra- and extra-cranial vessels, including: (1) the branches of the external carotid artery, the cavernous segment of the internal carotid artery, and ophthalmic artery, and (2) the dural anastomosis connecting the distal branches of the middle and occipital meningeal arteries with cerebral arteries. Additionally, collaterals between intracranial arteries include: (1) *basal collaterals* involving communication through the vessels forming the circle of Willis (circulus arteriosus), (2) *superficial collaterals* consisting of pial communication through the leptomeningeal artery, and (3) *intraparenchymal collaterals* encompassing small precapillary anastomoses between the branches of the perforating arteries [4].

Anatomical investigations have unveiled considerable interindividual variability regarding the completeness of the circle of Willis. A comprehensive survey utilizing magnetic resonance angiography demonstrated that a complete circle of Willis (type I) was present in only 25% of cases. Incomplete anterior half (type II) was observed in 57% of cases, while incomplete posterior half (type III) was identified in 3% of cases. Furthermore, incomplete anterior and posterior portions (type IV) were noted in 15% of cases. In terms of stroke severity on admission and discharge, it was observed that cases categorized as type I exhibited the lowest severity compared to types II and IV. Moreover, the presence of a complete circle (type I) was established as a significant predictor of a favorable outcome [5].

### 2.2. The Role of Posterior Circulation

Despite contributing only approximately one third of the overall cerebral blood flow, the posterior cerebral circulation remains responsible for numerous vital functions within the nervous system. It arises from the paired vertebral arteries and a singular basilar artery to provide blood supply to the inferior thalamus, temporal and occipital lobes, cerebellum, and brainstem.

The vascular reserve within the vertebro-basilar circulation includes bidirectional flow through the anterior inferior cerebellar artery, posterior inferior cerebellar artery), and cerebellar leptomeningeal vessels [6]. 

The leptomeningeal connections between the arteries of the cerebellum resemble those found in the cerebral pial network. These connections can reverse the blood flow through the tributaries of the basilar artery. Blood flow direction can be reversed due to hemodynamic connections involving the posterior communicating artery (PComA), the first segment of the posterior cerebral artery (PCA), and the carotid circulation. In cases of basilar artery occlusion, the PComAs reverse the blood flow through the basilar artery’s bifurcation, the PCA, and the superior cerebellar artery (SCA) located in the quadrigeminal plate [7]. Studies have shown that an increase in the size of the PComA vessel lumen is directly associated with better patient outcomes following occlusions of the basilar artery and the first segment of the PCA [8].

### 2.3. Collateral Circulation Risk Factors and Genetic Liability

The extent of collaterals at baseline seems to be dependent on genetic factors [9]. Aging plays an important role too, as several studies have shown a rarefaction of cerebral arterioles, a decrease in capillary density and a decrease in the number of venules and arteriole-to-arteriole anastomoses in aged humans and animal models [10]. It was observed that the rarefaction of collaterals leads to a failure of collaterals recruitment with an impaired pial vessel dynamic (a reduced red blood cell velocity) which worsens tissue perfusion [11,12].

Moreover, with age progression, there appears to be a decrease in vessel diameter, an increased vessel tortuosity, and, consequently, increased vascular resistances and larger infarct volumes. Aging also seems to inhibit vascular remodeling. These findings may be related to a decreased expression of VEGF and to a reduced synthesis of NO [13].

Sex differences also seem to play a role: it has been observed that women have a better collateral status compared to men, but the reasons for a such difference remain unclear. Studies in animals have found no difference between male and female subjects in terms of structure and function for what concerns the leptomeningeal anastomoses and the circle of Willis [14]. Nevertheless, women were found to have higher mortality and longer hospitalizations after a stroke compared to men. Vascular risk factors, such as diabetes and hypertension, have been indicated as responsible for impaired cerebral autoregulation and have been associated with larger infarct volumes [15]. Epidemiological studies have shown that women have a higher prevalence of hypertension and atrial fibrillation, but a lower prevalence of heart disease, peripheral vascular disease, smoking, and alcohol use [16]. Hypertension was indeed found to be associated with poor collateral status and little salvageable tissue [17]. Smoking, chronic kidney disease, and metabolic syndrome have also been associated with poor collaterals [18].

### 2.4. The Role of Cerebral Hypoperfusion

The normal cerebral blood flow (CBF) is typically around 50–60 mL/100 g/min. However, during hypoperfusion, when the CBF drops to a range of 10 to 20 mL/100 g/min, cellular functions are interrupted, but the cells remain structurally intact. Once the CBF falls below this threshold, the cells begin to undergo death within a matter of minutes [19]. Consequently, collaterals assume a vital role in substituting for the main arteries when they fail to deliver an adequate blood supply and maintain sufficient levels of CBF and cerebral blood volume (CBV). Such circumstances may arise from sudden occlusion of the blood vessels or as a consequence of chronic blood flow reduction. In cases where arterial occlusion abruptly halts blood flow through the vessel, an “early activation of collaterals” phenomenon is observed. On the other hand, if blood flow gradually and slowly diminishes, a delayed recruitment of collaterals occurs. The early activation primarily involves the polygon of Willis, while the late recruitment involves leptomeningeal vessels, intraparenchymal vessels, and branches of the external carotid artery such as the ophthalmic artery [20].

Collateral involvement arises due to the creation of a pressure gradient resulting from the obstruction of the main arteries, which diverts blood into the anastomotic vessels [21]. Moreover, the acidotic and hypercapnic environment generated by local hypoperfusion stimulates vasodilation, thereby increasing blood flow towards the ischemic region. Studies conducted on rat models have demonstrated that following a minor stroke, angiogenesis occurs at the periphery of the ischemic area, and a peak in cell proliferation is observed within the initial 48–72 h post-stroke [22,23].

### 2.5. Collateral Circulation and Neuromodulation

Many growth factors are indeed upregulated after ischemia: TGF-alfa, TGF-beta- TNF-alfa- IL-8, and, most importantly, VEGF. However, there are some doubts that the newly formed vessels are truly functional because, even though these molecular mechanisms lead to the formation of new microvessels, this process seems to be aimed at macrophage infiltration and at the removal of cellular debris from the necrotic tissue. In addition to this, the newly formed microvessels are maintained for a limited time only and their regression can be observed already after 30 days since reperfusion. In any case, data seem to indicate that they are beneficial to neurovascular repair [24,25].

In contrast to angiogenesis, which is the formation of new capillaries by sprouting or “de novo” and is triggered by oxygen deprivation, arteriogenesis is the remodeling of existing arterio-arteriolar anastomoses toward fully functional arteries.

Arteriogenesis is initiated by fluid shear stress (FSS) following sudden occlusion or progressive stenosis in an arterial vessel. This process encompasses various interconnected events, including endothelial activation, monocytic cell recruitment, promotion of inflammatory processes, secretion of growth factors and cytokines, extracellular matrix digestion, and proliferation of smooth muscle cells, all contributing to the outward remodeling of collateral vessels [26].

FSS refers to the tangential frictional forces exerted by the flow of blood, that is a viscous fluid, against the vessel wall. Arterial occlusion leads to a decrease in distal vascular pressure, resulting in increased flow through pre-existing collaterals due to the pressure gradient. Initially, the increased FSS on the endothelium triggers a cascade of signaling events. As the collaterals grow in diameter, FSS decreases, and collateral flow decreases, establishing a self-regulating mechanism.

Endothelial cells lining the blood vessel lumen have the ability to sense changes in flow shear stress and transmit signals to smooth muscle cells in the tunica media and fibroblasts in the tunica adventitia. Microarray data have identified a potential role for the shear stress-sensitive gene transient receptor potential cation channel subfamily V member 4 (Trpv4), a calcium channel found in endothelial cells. Pharmacological activation of Trpv4 has been shown to strongly enhance cerebral arteriogenesis and collateral flow, while blocking Trpv channels reduces collateral flow [27,28].

The signal generated by FSS is likely transmitted through nitric oxide (NO), which can diffuse through the basement membrane separating the endothelial cells from the smooth muscle cells and can interact with prostacyclin.

Although the initiation of arteriogenesis does not necessarily require the presence of hypoxia-inducible factors (HIFs), hypoxia activates signaling pathways that lead to the production of various molecules, including VEGF and inflammatory cytokines. HIF-1 also promotes the secretion of integrin β2, which, in turn, induces the adhesion and infiltration of leukocytes and vascular progenitor cells into hypoxic sites [29,30].

Recent studies have highlighted the role of different subsets of innate and adaptive immune cells, such as monocytes/macrophages, T helper 17 cells, regulatory T lymphocytes, and natural killer cells, as regulators of arteriogenesis. There is growing evidence supporting the critical involvement of monocytes and monocyte signaling in arteriogenesis. Depletion of monocytes has been shown to impair arteriogenesis in animal models of hindlimb ischemia. M1-type macrophages express inducible NO synthase (iNOS) and proinflammatory cytokines like IL-1 and IL-12, while the M2-subset expresses arginase 1, the anti-inflammatory cytokine IL-10, and VEGF [31,32].

The deletion of one allele of the Phd2 gene not only induces a shift in macrophages towards an M2 phenotype but also enhances arteriogenesis following hindlimb ischemia. These infiltrating macrophages play a crucial role in producing numerous angiogenic growth factors, including MCP-1, VEGF, FGF, GM-CSF, HGF, TNF-α, TGF-β, and PDGF, which are essential for the development and maturation of collaterals. Arteriogenesis shares certain similarities with an active inflammatory process. However, it remains uncertain whether a specific type of inflammatory response is necessary for arteriogenesis or if the inflammatory process triggered by the ischemic insult, along with FSS and the gene products released by activated endothelium, is sufficient for collateral vessel formation. During arteriogenesis, the infiltration of monocytes and macrophages is followed by collateral growth and enlargement, which require the proliferation of endothelial and mural cells, as well as the remodeling of the extracellular matrix. Matrix metalloproteinases (MMPs) and their inhibitors, tissue inhibitor of metalloproteinases (TIMPs), play a crucial role in the turnover and remodeling of the extracellular matrix.

Previous research has shown that during collateral remodeling, there is an upregulation of MMP-2, MMP-9, and TIMP-1 in the intima, indicating that the balance between these MMPs and TIMPs is crucial for the maintenance and remodeling of the vessel wall [33].

For instance, in a murine hindlimb ischemia model, the Lepr-db/db mutation hindered the ischemia-induced upregulation of MMP-2, MMP-12, and MMP-16, leading to impaired arteriogenesis. On the other hand, patients with Moyamoya disease, a chronic cerebrovascular condition characterized by pathological instability in the vessel wall and abnormal collateral vessel growth, exhibited significantly elevated levels of MMP-9 in their serum, but not MMP-2 [34].

The process of arteriogenesis takes place in the context of the leptomeningeal anastomoses which lie in an oxygen-rich environment and it has been associated with prior episodes of symptomatic or asymptomatic ischemic stress in the affected territory, something that occurs in the setting of atherosclerotic diseases rather than of a cardioembolic event [35].

Therefore, the so-called cerebral ischemic preconditioning refers to transient, sublethal ischemic insults which make the brain more tolerant to subsequent, otherwise lethal, ischemic events. Several studies have demonstrated that it involves molecular mechanisms including the activation of NMDA and adenosine receptors, MAPK signaling, heat-shock proteins, and NO upregulation. Moreover, preconditioning seems to be capable of increasing collateral anastomoses and enlarging the diameter of leptomeningeal vessels in rat models [36].

What matters in the setting of ischemia is preventing the collapse of collaterals in order to maintain an adequate tissue perfusion, something that was achieved in studies on remote ischemic preconditioning and that may hopefully be applied in the clinical setting as well [37].

### 2.6. The Role of the “Core” and the “Penumbra”

In recent years the collateral circulation has received much attention from researchers because it has been shown to be related to the outcome of patients with stroke.

Stroke patients with diminished collaterals have a rate of worsening four times higher and a better collateral status can be considered an independent predictor of an improved functional outcome [38]. A good collateral circulation was associated also with better functional recovery at 90 days [39]. The importance of the brain collateral circulation arises from the core/penumbra concept: the observation that, in the stroke setting, it is possible to observe a gradient of hypoperfusion, ranging from an area of severe hypoperfusion to the most peripherical regions where hypoperfusion is mild. One may thus imagine three concentric zones: a “core” where hypoperfusion is severe, the “penumbra” where neural cells are dysfunctional but still salvageable if adequate perfusion is restored, and the “oligemia” where hypoperfusion is mild. Another important observation was that the tissue under severe hypoperfusion survives a shorter period of time and that the tissue in “penumbra” will inevitably progress to irreversible damage, which is to say, the “core” grows over time if perfusion is not restored [25]. What determines the survival of the penumbra is indeed the collateral circulation, but it cannot save the penumbra indefinitely [40].

An inverse correlation exists between total infarct volume, collateral grade (a measure of the extent of vascularity as measured at the Sylvian fissure and the cerebral convexity), and clinical outcome (expressed as modified Rankin scale at discharge) and because of the different vascular arrangement, collaterals will protect the cortex more than the deep tissues. Every 10 mL increase in pretreatment ischemic core volume reduces the odds of favorable functional outcomes by 20–30% [41].

The observation that the core grows over time if perfusion is not restored had a dramatic clinical impact and allowed trials to be run which ultimately led to the revascularization procedures: IVT and MT which are the only therapeutic strategies that can alter the prognosis of stroke patients. The impact on functional outcomes is directly related to the timing of the intervention, resulting in both greater treatment efficacy and a reduction in complications associated with these therapies, including hemorrhagic infarction. Indeed, the most recent guidelines recommend performing thrombolysis within 4.5 h without advanced imaging (CTP) or within 9 h with CT perfusion and to perform thrombectomy within 6 h without CTP or within 24 h if CTP is available [42]. Despite proper and timely administration of these therapies, the functional outcome is not always optimal.

Multiple studies have acknowledged the significant role of collateral circulation in determining prognosis in stroke patients. This role proves valuable both in the diagnostic phase, aiding in the selection of optimal therapeutic approaches, and in the post-revascularization phase, serving as a prognostic indicator. One study suggests that collateral circulation status should be considered alongside symptom onset time when making therapeutic decisions, particularly for thrombolysis. Specifically, this study reveals that the presence of a robust collateral circulation is associated with better Rankin scores, regardless of the time elapsed between symptom onset and recanalization. Patients with good collateral circulation exhibited favorable clinical outcomes (mRS 0–2) when treated within 345 min of symptom onset, while patients with poor collateral status showed good outcomes when treated within 172 min of stroke onset [3].

The most crucial factor in therapy selection for stroke patients is the ASPECT (Alberta Stroke Program Early CT) score, which represents the extent of infarct expansion observed in CT brain imaging and serves as a prognostic determinant of ischemic stroke. A low ASPECT score, indicative of a large ischemic lesion, is an absolute contraindication for both thrombolysis and thrombectomy. However, several studies suggest that including collateral imaging in CT angiography (CTA) along with the ASPECT score can enhance prognostic accuracy and improve therapeutic decision-making [5]. In fact, a study examined the prognostic value of collateral circulation in patients with low ASPECT scores who underwent thrombolysis, demonstrating that despite a large infarction area, the presence of good collateral circulation was associated with better functional outcomes [43]. A meta-analysis of 42 studies also revealed that better collateral status prior to treatment correlated with improved functional outcomes at 3–6 months in patients treated with thrombolysis, and this association appeared more significant in patients treated within extended time windows [44].

During the administration of reperfusion treatments, collateral circulation status can provide crucial prognostic information. Post hoc analysis of the interventional management of stroke III trial, a multicenter phase 3 trial involving 900 subjects evaluating low-dose IV tPA plus thrombectomy or IV tPA alone (which was discontinued due to futility), demonstrated the importance of collateral circulation status for clinical outcomes. Specifically, patients with better collateral status exhibited higher reperfusion and recanalization rates compared to those with poorer collateral status.

Collateral circulation status also appears to be associated with intraoperative mortality and hemorrhagic infarcts. In a study of 246 subjects, including 205 with good collaterals and 41 with poor collaterals, mortality was 41% in patients with poor collaterals compared to 12% in patients with good collateral status. Hemorrhage occurred more frequently in patients with poor collateral circulation (15% vs. 4.9%) [45]. Vessel hyperintensity in FLAIR sequences serves as an index of collateralization degree in patients with occlusion of large intracranial arterial axes. This index is associated with smaller lesion growth and better functional recovery [46]. A subanalysis of the WAKE UP study dataset demonstrates that a lower number of vascular hyperintensities in FLAIR is associated with better functional outcomes in patients treated with thrombolysis. PWI sequences also provide indices of patency and collateral status, which predict complete recanalization after thrombolysis and good functional outcomes [47].

## 3. Imaging

Digital subtraction angiography (DSA) has been widely acknowledged as the reference standard for assessing collateral circulation, although CTA-based scores are more commonly employed. The evaluation of collateral circulation status can be performed using quantitative or qualitative methods. The quantitative scoring involves the utilization of Stroke Viewer software, which calculates the ratio between the affected hemisphere and the contralateral hemisphere. Initially, the software identifies the ischemic hemisphere through an occlusion detection algorithm, followed by the segmentation of vessels distal to the occlusion [48].

The quantitative score, in technical terms, is calculated as the ratio of the volume of collaterals in the occluded area to the volume of vessels in the corresponding area on the unaffected side. The vessel volume is determined through manual segmentation of CTA images using the open-source software ITK-SNAP.

The vCS (visual collateral score) is instead divided into four points according to the proportion of collaterals in the occluded territory: 0 = absent collaterals (0%), 1 = poor collateral circulation (≤50%), 2 = moderate collateral circulation (50–100%, not including 100%), and 3 = good collateral circulation (100%). In predicting clinical outcome, qCS is superior to vCS as it has a higher AUC value and a higher sensitivity [49].

### 3.1. Manual Assessment

The most widely recognized grading system is the American Society of Interventional and Therapeutic Neuroradiology/Society of Interventional Radiology (ASITN/SIR) collateral scale based on DSA, which assesses the cerebral collateral status as follows: grade 0: no collaterals to the ischemic site are visible;grade 1: slow collaterals to the periphery of the ischemic site with persistence of some of the defect;grade 2: rapid collaterals to the periphery of ischemic site with persistence of some of the defect and to a portion of the ischemic territory;grade 3: collaterals with slow but complete angiographic blood flow to the ischemic bed in the early venous phase;grade 4: collaterals with rapid and complete angiographic blood flow to the ischemic bed.

Grades 0–1, 2, and 3–4 are usually regarded as poor, moderate, and good collateral flows [50].

The most important collateral grading methods based on CTA are:
the method proposed by Miteff et al.: grading collateral flow distal to MCA occlusion.

The grading of collateral status is categorized into three groups: good (indicating reconstitution of major MCA branches distal to the occlusion), moderate (demonstrating the presence of some MCA branches in the Sylvian fissure), or poor (with only distal superficial MCA branches reconstituted);

the method proposed by Maas et al.: assessing collaterals at the Sylvian sulcus and cerebral convexity as well as the collateral pathways via the circle of Willis;the method proposed by Tan et al.: grading collaterals in the MCA territory;the regional leptomeningeal collateral (rLMC) score: assessing collaterals in MCA cortical regions, parasagittal ACA territory, the basal ganglia and the Sylvian sulcus.

Overall, the scores by Maas and Miteff perform best in terms of functional outcome prediction. Given the high predictive value and high sensitivity (96%) of the Maas Score, it seems to be the most suitable for decision-making in patients with AIS. Patients with a Maas Score of two have the potential to profit from thrombectomy, while patients with a score of one have virtually no potential to achieve good outcome. Some of the features provided by computed tomography perfusion including the mismatch ratio and the infarct core, are of major importance in this regard. The mean rCBV (relative CBV) showed a high and statistically significant correlation with functional outcome (Figure 1). The Miteff Score correlates best with the mean rCBV, followed by the Maas Score, which reflects their good usefulness prediction of the functional outcome [51].

In a study conducted by Yeo et al., it was demonstrated that, among the various grading systems, only the Miteff grading system exhibited reliability in identifying both favorable and poor outcomes in patients with anterior circulation acute ischemic stroke treated with IVT. Other grading systems, on the other hand, were only able to predict poor outcomes [52]. Although there have been studies comparing the clinical significance of these grading methods, the results have been inconsistent, and none of the collateral grading systems have been thoroughly validated in large-scale studies.

### 3.2. Automatic Assessment

A recently introduced technique for assessing collateral status is a color-coded mapping method called “ColorViz” (GE Healthcare Fast Stroke). This post-processing tool enables rapid and clear evaluation by maintaining the temporal resolution of multiphase CTA (mCTA) and generating a single image that combines three different cerebral vascular phases using a time variant color map. The color-coded mCTA summation maps are generated using the Fast Stroke software (GE Healthcare, Milwaukee, WI, USA) on a workstation. This software processes the complete set of CT stroke protocol images to create a single color-coded map known as ColorViz. The vessels depicted on the map are assigned different colors based on the arrival time of the contrast medium using a per-person adaptive threshold technique. Specifically, the vessels are assigned the following colors: (a) red (indicating maximal enhancement during the arterial phase), (b) green (representing the early venous phase), and (c) blue (representing the late venous phase). The post-processing procedure is straightforward and fully automated.

A score of three (“good”) is given when a well-preserved or slightly reduced representation of the collateral circles is observed on the affected side, with the predominant color being red. A score of two (“intermediate”) is assigned when a similar or reduced extension of collateral circulation is observed compared to the healthy side, and the predominant color is green (indicating slightly slowed circulation). Lastly, a score of one (“poor”) is assigned when the most prevalent color is blue (indicating significantly slowed circulation compared to the contralateral side) or when the extension of collateral circles is markedly reduced, even in the presence of a few green or red vessels (Figure 2) [53].

Another software for automatic detection of collateral circle status is e-CTA from Brainomix. This software is based on a convolutional neural network structure that allows highlighting asymmetries between the vessels of the left and right hemispheres of the brain. Then it provides heat maps and scores for the percentage of vessel density as output (Figure 3A,B) [54].

## 4. Clinical Implications

The status of collateral circulation may provide important prognostic information also during reperfusion procedures. Post hoc analysis of the interventional management of stroke III trial (a multicenter phase 3 trial of 900 subjects evaluating treatment with low-dose IV tPA plus thrombectomy or IV tPA alone) highlighted the crucial role of collateral circulation status for clinical outcome. Specifically, reperfusion and recanalization rates of treated patients were higher in patients with higher collateral status. The centrale role of collateral circulation status also appears to be related to intraoperative mortality and hemorrhagic infarcts. In a study of 246 subjects, including 205 with good collaterals and 41 with poor collaterals, mortality was 41% in patients with poor collaterals compared to 12% in patients with a good collateral status. Hemorrhage occurred more frequently in patients with poor collaterals (15% vs. 4.9%) [55].

According to an analysis of the DEFUSE 3 cohort, it was found that, although a good collateral status was associated with smaller infarct volumes, it did not serve as a reliable predictor of the clinical outcome [56].

One possible explanation for this observation could be that certain specific brain regions have a more significant impact on functional independence. For example, lesions involving the central sulcus and/or the subcortical white matter containing corticospinal tracts are expected to result in greater disability. Successful reperfusion tends to preserve the pre-central gyrus and the posterior limb of the internal capsule, which are regions containing corticospinal fibers. The vulnerability of the lentiform nuclei may be due to their blood supply from the lateral lenticulostriate arterial branches of the MCA, which lack collateral support. Moreover, these areas may be relatively more susceptible to hypoperfusion, thus a favorable collateral status at baseline may not be adequate to sustain the viability of the lentiform nuclei until endovascular reperfusion is achieved. Infarction in the basal ganglia and insular cortex is associated with a worse outcome in patients with anterior circulation LVO (large vessel occlusion).

Patients who achieved successful reperfusion showed smaller final infarct volumes (*p* < 0.0001), lower discharge mRS scores (*p* < 0.0001), and lower rates of disability/death (*p* < 0.0001). Patients with poor collateral status had a shorter onset-to-catheterization time (*p* = 0.023 on post hoc analysis), larger infarct volumes (*p* < 0.0001), worse discharge mRS (*p* = 0.007), and higher rates of disability/death (*p* = 0.009). However, the statistical significance was not maintained after adjusting the *p*-value threshold to <0.003 with Bonferroni correction for multiple comparisons.

While higher mTICI (modified treatment in cerebral ischemia) scores may specifically preserve the deep white matter tracts in the MCA territory and the posterior limb of the internal capsule, the baseline collateral status may instead specifically determine the fate of MCA border zones with ACA and PCA territories. Both the collateral status and the mTICI score were independently associated with final infarct volume after adjusting for age, National Institutes of Health Stroke Scale (NIHSS) score at admission, MT, and occlusion side [57].

In a post hoc analysis of the DAWN trial conducted by Liebeskind et al., it was observed that patients with good collaterals, who received endovascular therapy during the late therapeutic window for acute ischemic stroke, exhibited several positive outcomes. These included smaller core infarcts upon presentation, slower progression of infarction, and higher rates of functional independence at 90 days. These findings align with previous research in the early time window of anterior circulation acute ischemic stroke, such as the SWIFT trial, where better collaterals were associated with increased likelihood of successful revascularization and favorable clinical outcomes [54].

In another study by Xu et al., a total of 219 patients were enrolled, with 109 classified in the poor collateral group and 110 in the good collateral group. The two groups had similar age and sex distributions. Patients with good collaterals, compared to those in the poor collateral group, exhibited a lower prevalence of hypertension, diabetes, and ischemic stroke. They also had higher rates of statin use, lower systolic blood pressure levels, and lower NIHSS scores upon admission. There were no significant differences between the groups in terms of history of atrial fibrillation, use of antiplatelet or anticoagulant medications, glucose levels upon hospital arrival, time from stroke onset to thrombolysis, time from stroke onset to groin puncture, and time from stroke onset to revascularization. The poor collateral group had a higher 90-day mortality rate (26.6% versus 20.0%) compared to the good collateral group. These findings support the notion that pretreatment collateral status in patients undergoing MT for ischemic stroke is associated with successful reperfusion and improved clinical outcomes [58].

In a single-center study conducted by Lagebrant et al., it was demonstrated that women who underwent MT for stroke had better collateral status compared to men, but paradoxically experienced worse outcomes. These differences remained significant even after accounting for age. Additionally, utilizing the collateral grading system developed by Tan et al., the study reported that a higher proportion of women (88%) exhibited moderate or good collateral circulation grades compared to men (80%) (n = 1290, *p* < 0.01). The exact reasons for these disparities in collateral flow between men and women undergoing MT remain unknown. However, it is acknowledged that factors such as smoking, alcohol use, large vessel atherosclerosis, and peripheral vascular disease are more prevalent among men with stroke than women. These factors contribute to early vessel aging and impaired autoregulation, which could potentially explain the lower collateral scores observed in men [59].

In a study by Guglielmi et al., a significant shift towards better collateral scores was observed in cases of stroke caused by cervical carotid atherosclerosis (adjusted common odds ratio: 1.67 [95% CI, 1.17–2.39]). When collateral scores were categorized into two levels, namely good (grade 2–3) and poor (grade 0–1), patients with cervical carotid atherosclerosis exhibited a significantly higher proportion of good collateral scores compared to those with cardioembolic stroke (130/184 [71%] versus 266/441 [60%], adjusted odds ratio: 1.84 [95% CI, 1.15–2.94]). Furthermore, patients with cervical carotid atherosclerotic stroke demonstrated a lower median modified Rankin Scale (mRS) at 90 days compared to patients with cardioembolic stroke. However, no statistically significant differences were found in the proportions of patients achieving an mRS score of 0–2 (46% versus 35%, adjusted odds ratio: 1.36 [95% CI, 0.90–2.07]) or in terms of mortality (23% versus 33%, adjusted odds ratio: 0.80 [95% CI, 0.48–1.34]) at 90 days. Consistent with their hypothesis, the researchers discovered that patients who underwent endovascular therapy (EVT) for anterior circulation LVO caused by cervical large-artery atherosclerosis displayed more extensive cerebral collateral circulation and better functional outcomes at 90 days compared to those with cardioembolic stroke. However, no statistically significant differences were observed in terms of functional independence (mRS score of 0–2) or mortality between the two groups [60,61].

## 5. Discussion

The collateral network comprises arterial vessels that provide support to cerebral circulation in hypoperfused areas affected by ischemic stroke. The establishment of this network is crucial for preserving the ischemic penumbra and salvaging viable tissue. Numerous studies in the recent literature have demonstrated a direct correlation between collateral circulation and both clinical and therapeutic outcomes in patients with ischemic stroke. In this review, we extensively analyzed the literature data to elucidate the impact of adequate collateral circulation on the success of reperfusion therapies in acute ischemic stroke and prognosis. Mortality data clearly favor patients with better collateral circulation over those with poor collateral status. The results of the study indicate higher reperfusion and recanalization rates in patients with superior collateral status.

To improve the accuracy of identifying hypoperfused areas amenable to salvage through collateral circulation, various sophisticated radiological software tools have been developed. One such tool is “ColorViz,” which provides immediate post-processing visualization of cerebral vessels using colored maps. The vessels enhancing maximally during the arterial phase are assigned the color red, while the early and late venous phases are represented by green and blue colors, respectively. The presence of well-represented collateral circles is indicated by a predominant red color in the maps. This graphical representation allows for easy interpretation and facilitates the assignment of a quantitative score to assess the extent of collateral circulation.

Despite this, we still do not have one or more standardized and reproducible software available that can be applied in clinical practice on all kinds of patients. In particular, a computer tool that allows us to accurately highlight the presence and extent of collateral circulation in a quantitative way is desirable; it could be useful to select those patients with onset of ischemic stroke symptoms beyond the classic time windows. This would allow a careful selection of patients to be treated, but above all to be “not treated”, as the latter are those at greater risk of complications and post-stroke mortality.

Although not yet a reality, it is foreseeable that, in the near future, the study of collateral circulation, along with the assessment of the ischemic penumbra, could serve as fundamental screening criteria for selecting appropriate therapeutic interventions in acute ischemic stroke. The presence, quality, and extent of collateral circulation could become determinants in choosing the optimal treatment approach during the acute phase, as well as prognostic indicators for survival, mortality, hemorrhagic transformation of ischemic lesions, and recovery of functional autonomy.

The future is unwritten. Probably, in the near future, collateral circulation detection will become the primary condition for the stroke treatment or not-treatment decision.

## Figures and Tables

**Figure 1 diagnostics-13-02425-f001:**
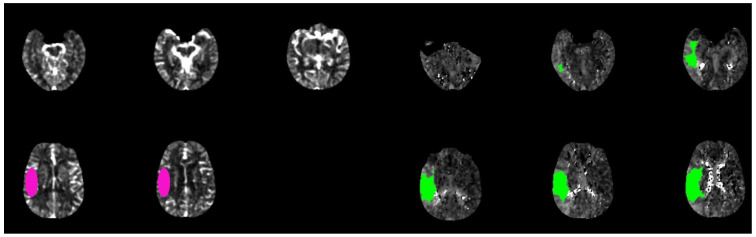
Brain CT-scan perfusion analyzed with rapid software in a patient with acute ischemic stroke show a favorable mismatch core (pink circle) vs. penumbra (green circle) with the following parameters: CBF < 30% volume: 15 mL; Tmax > 6.0 s volume: 160 mL; mismatch volume: 145 mL; mismatch ratio: 10, 6.

**Figure 2 diagnostics-13-02425-f002:**
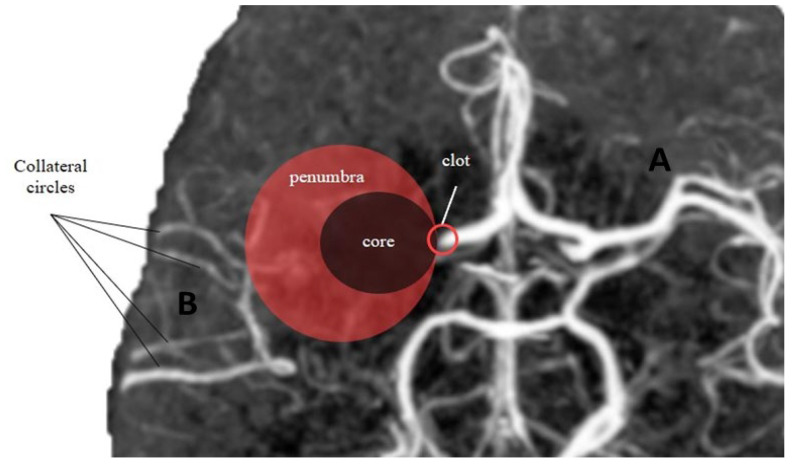
Core-penumbra ratio under different collateral circles condition. (**A**) Good collateral circles result in a large core-penumbra mismatch. (**B**) Poor collateral circles lead to a big core with small core-penumbra mismatch.

**Figure 3 diagnostics-13-02425-f003:**
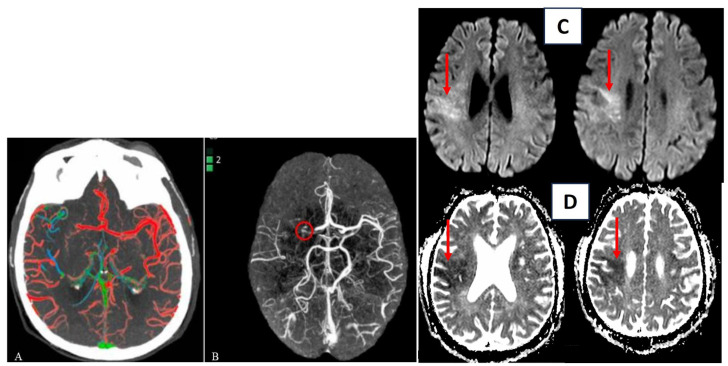
Automatic evaluation of collateral circles by ColorViz (**A**) and e-CTA (**B**) (reproduced under the terms and conditions of the Creative Commons Attribution (CC BY) license from Verzolotti et al. (**A**) and Zelenak et al. (**B**)). Corresponding acute ischemic lesion in the area supplied by the right middle cerebral artery in DWI sequences (red arrow (**C**)) and ADC sequences (red arrow (**D**)).

## Data Availability

Not applicable.

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
