# Peer review of "The Pathophysiology of Collateral Circulation in Acute Ischemic Stroke"

_diagnostics, 2023, doi:10.3390/diagnostics13142425_

Round 1
Reviewer 1 Report
The title indicates adequately the study design.
The abstract provides an informative and balanced summary of the study.
The manuscript is well structured with an adequate and clear presentation of scientific review of the literature.
The cited references contains recent publications.
The discussion is fluid, and the passages are clear.
Images are well explained but insufficient. The DWI/ADC sequences and especially PWI images should be added.
The manuscript is fluid and clear. Minor editing of English language is required.
Author Response
Dear reviewer,
thank you for revision. As requested I add more images in this paper, DWI/ADC and TC perfusion images.

Reviewer 2 Report
The review provides a comprehensive overview on pathophysiology, clinical and therapeutic aspects of collateral cerebral circulation in acute ischemic stroke. The topic is of interest, since collateral flow is an important factor to be considered when chosing the best treatment option for patients with acute stroke and harbors relevant prognostic implications. The manuscript is globally well written and only few minor edits are required in my opinion:
- please better describe in the introduction current gold standards on extended reperfusion time window for both IVT and EVT. It could be worth to briefly expand citing controversies and reasons hampering the effectiveness of appropriately administered reperfusion treatments in some cases (beyond poor collateral circulation).
- I suggest to expand the discussion section by providing authors' personal point of view of future challenges and best imaging modality to improve the selection of patients for reperfusion therapies taking into more account the state of collateral circulation in treament decisions.
